# The Development of Emotional Programmes in Education Settings during the Last Decade

**DOI:** 10.3390/children9040456

**Published:** 2022-03-24

**Authors:** Beatriz Muñoz-Oliver, Pedro Gil-Madrona, José Luis Gómez-Ramos

**Affiliations:** 1Department of Didactics, Physical, and Music Education of Physical Education, Faculty of Education at Albacete, University of Castilla-La Mancha, 02071 Ciudad Real, Spain; bmunoz@edu.jccm.es; 2Department of Pedagogy, University of Castilla-La Mancha, 02071 Ciudad Real, Spain; joseluis.gomez@uclm.es

**Keywords:** emotional intelligence, education, assessment, emotional programmes

## Abstract

Within the psychological domain of emotional intelligence, experimentation on emotional education programmes in school contexts constitutes one of the most compelling research lines in recent years. On this basis, this study presents a review of forty-one programmes implemented in educational settings. The results obtained from the primary and secondary scrutinised sources show the need to integrate families into programme interventions and the importance of teacher training in socio-emotional competencies. Likewise, the importance of interconnecting natural educational settings with research activity is considered a fundamental aspect in designing, implementing, and evaluating such programmes. Thus, the present study aims to represent these programmes’ characteristics, evaluation, and results for the ulterior development of specific and contextualised proposals.

## 1. Introduction

Emotional intelligence (EI) represents the ability of individuals to reason and employ self-emotions for the enrichment of thoughts and knowledge [1,2,3]. It is probably one of the most current issues in the field of Education. In addition, it has a positive reception in other sectors, such as clinical, business and production, commercial and advertising, and tourism. However, this has not always been the case. To date, people’s cognitive skills have been overvalued over the other aspects, such as emotional and social ones. Due to the benefits reported to educators and learners, emotional intelligence (EI) is an issue of consideration in many educational settings [4,5]. Notwithstanding the differences in the results obtained from the varied research carried out, EI is said to predict necessary outcomes [6]. Due to the variances formed around the concept, we consider it convenient to address the problem of the theoretical nature from which the EI models start to identify those best fitting the initial education environments. For example, instead of models based on the processing of emotional information focused on emotional skills and personality attributes [7,8], we consider that skills-based models [3] might be more appropriate for the initial stages in education.

Though there is an abundant number of bibliographies on emotional intelligence, there is an apparent lack of specific literature on the training and guidance of pedagogical models for educators to later design, implement, and assess emotionally based learning environments. In this regard, the (long-term) assessment and transfer of EI skills in scholars remain crucial since the time variable also positively or negatively influences the retention of the information [9]. Thus, the EI is a genuine skill based on the adaptive use of emotions to solve the problems that emerge from the different school environments and learning conditions from the theoretical foundation presented. Even though the implementation of adapted teaching models to promote emotional skills in learners is included in the curriculums of education [10], there is a lack of didactic guidance serving as a referent for the efficient encouragement of emotional education (EE). According to Ibarrola [11], education systems are deficient unless emotional education is appropriately incorporated into them. According to this author, much of the learning problems faced by students originate from emotional issues, not a lack of skills, which is a common perception.

To verify the nature of these programmes concerning their implementation in early childhood education, in this study, we present an overview of 41 investigations in which emotional education programmes developed and evaluated in the last 10 years are identified in Early Childhood Education, Primary Education, Secondary Education, and University Education. The bibliographic review is mainly based on primary and secondary sources. The descriptors considered in the search are the different theoretical models of IE, their dimensions, and their working and development approach. Thus, the present bibliographic review aims to discern the characteristics of the IE programme’s implementation and its evaluation, as well as the findings and aspects of the research activity, to offer work guidelines that are specific for future plans and research concerning emotional issues in early childhood education. It can be observed from the analysis of the results obtained from the analysed programmes that there is little participation from the families, little teacher training in socio-emotional competencies, and a disconnect between educational contexts and research activity (Table 1).

There is a hypothesis that behavioural and cognitive-behavioural-oriented programmes are more systematic than those not revealing such facets. Additionally, they unveil more straightforward goals and, consequently, are more effective. Based on the systematic presentation in Table 1 and an analysis of the result patterns, this study is characteristic because it considers programmes whose researchers themselves implement the activities integrated in each of the considered programmes. The meta-analyses carried out, to date, seem to indicate that the most effective programmes are theoretically coherent and have highly interactive demands. Those who use various teaching methods implemented through small groups, work on life skills, develop environmental strategies, and are integrated into the community are also seen as efficient [12].

## 2. Conceptualisation and Core Models of Emotional Intelligence

The genesis of the study of EI was in 1940, originating from David Wechsler’s idea on the ‘general intelligence of non-intellectual aspects’ in the last century. In 1948, Leeper established that emotional thinking was part of and contributed to logical thinking and intelligence in general. One of the most controversial aspects of this concept relies on the theoretical nature from which the EI models started. In this regard, it is possible to establish two distinctions between IE models: one based on the processing of emotional information and focused on basic emotional skills; another based on mixed models considering personality traits [7,8,13]. On the one hand, the best model representing the first category is the skills-based EI Model of Mayer and Salovey [3]. From this theoretical framework, EI is conceived as a genuine intelligence based on the adaptive use of emotions to solve problems by effectively adapting themselves to the environment by which they are surrounded. On the other hand, the vision of mixed models, such as those of Bar-On [14] or Goleman’s [1], are more extensive and somewhat vague. These models focus on stable behavioural traits and the personality variables, such as empathy, assertiveness, and impulsivity [15].

To broaden the vision of the factors included in EI, Bar-On [14] proposed a multifactorial model based on Emotional Quotient (EQ-i) from a multifactorial and eclectic approach aiming to achieve an operational definition and quantitative description of EI. The model focused on accurately answering the question of why some people could succeed in life while others could not. In the clinical psychology field, experiences have shown that the key to determining and predicting success is not only a matter of cognitive intelligence, since some people with high intelligence are not successful in life and others with more limited intelligence achieve somewhat greater success. Since the fundamentals might be of interest in the field of education, the Bar-On inventory was examined worldwide via longitudinal and cross-sectional studies.

Apropos of conceptualisation, EI was conceptually defined by Salovey and Mayer as social intelligence encompassing the abilities to monitor and understand the emotions of the self and others [3]. Therefore, the training of socio-emotional skills for students is becoming necessary, and most teachers consider these skills to be essential [7,12,15,16]. According to Ibarrola [11], the education system is lame, as long as it does not incorporate EE, stating that many academic performance problems have their origin in emotional issues and not in a lack of skill. Additionally, it is increasingly evident that general success and well-being in adulthood may result from early childhood learning by using these social and emotional skills to productively cope with life changes, as they reduce the risk of suffering as a result of mental health problems and improve psychological well-being [17]. Within the clinical psychology framework, experiences showed that the key to determining and predicting success was not only a matter of cognitive intelligence, since some people with supposedly high intelligence were not successful in life, while others with less intelligence obtained much more outstanding achievements.

Schools are the ideal place for the promotion of EI [1,17]. In this sense, Gardner’s *Theory of Multiple Intelligences* (1983) has been considered as the most influential work in the field of Education. According to this author, intelligence changes and develops, based on the individual’s experiences throughout his or her life, resulting from the interaction between biological and environmental factors. For Gardner [18], humans are better defined by saying that they have a series of relatively independent bits of intelligence than by establishing a single intelligence defined by an IQ. More specifically, he affirms that there are many ways to be intelligent. Despite the criticism it received, it is inevitable that Gardner’s theory highlighted inter-individual variability in the classroom and that there must be different teaching methods adapted to that form of education. Another advantage of Gardner’s view is that it has many direct educational practice applications, advocating a teaching–learning process that develops learners’ intelligence. Thus, Gardner’s *Theory of Multiple Intelligences* is the basis for creating and applying novel, motivating, integrative, and creative strategies, so that students, in their leading role, build broad knowledge schemes. Compared to others, this perspective goes beyond everyday knowledge limits, bringing them closer to understanding and creative potential by developing or activating other bits of intelligence [19]. Table 2 displays the basic models of emotional competencies and their components.

## 3. Emotional Intelligence in Education Settings

Socio-emotional skills deficiencies affect students inside and outside the school context [15]. There are four fundamental areas in which a lack of EI causes or facilitates behaviour problems among students: interpersonal relationships, psychological well-being, academic performance, and disruptive behaviours. According to the review carried out by Zeidner et al. [19], some beneficial effects of emotional skills work stand out, such as the regulation of emotions and a positive attitude, and the reduction in counterproductive or distorting behaviours, such as stress or anxiety. Other studies reveal the benefits of having high EI levels. In this regard, an improvement to psychological well-being influences the mental health of school children, allowing them to acquire an emotional balance to enhance their academic performance [15,17].

Being educated at school, in terms of emotions, should be considered a curricular priority. EE programmes can reduce maladaptive behaviours and favour general well-being at intrapersonal and interpersonal levels; at an intrapersonal level, providing strategies for emotional and social regulation, and, at an interpersonal level, increasing strategies to manage the emotions of others [15]. Thus, the aspects, such as self-esteem, can significantly increase the well-being of students, while, at the same time, allowing them to improve the performance and climate in the classrooms. Receiving an education in emotional skills helps students to become aware of their emotions and those of others. Additionally, it enables them to understand and regulate themselves, has beneficial effects on the physical and psychological health of adolescents on multiple levels, and positively impacts their academic performance [8]. Above all, the development of socio-emotional competencies in adolescence, which has repercussions in adult life, is a nonspecific prevention formula for drug use, stress, anxiety, and violence, thus reducing vulnerability and increasing self-esteem and skills that allow individuals to adopt a positive attitude towards life [16,20].

## 4. Goal and Objectives

This study aims to identify the EE programmes that were developed and evaluated in the last ten years for early childhood education, primary education, compulsory secondary education, and university environments. The idea is to understand the characteristics of these programmes, their evaluation, and the findings and aspects of the research activity. Some objectives emerge from the goals, such as identifying the educational stage at which the programme is implemented, the author of the evaluation, the place of intervention, and the extraction and synthesis of the programmes’ characteristics and their evaluation (namely, previous teacher training; the format of interventions according to the participation of teachers from the centre and external personnel; the involvement of families in the programmes and the evaluation; the duration of the programmes in lessons and months; the research design; the sample size; and the dependent variables with significant outcomes after the intervention).

## 5. Methods

The evaluation of the systematic review considers the references from the last decade. Following the classification established by Guirao-Goris et al. [21], primary sources were used (books, papers on theoretical approaches, research, and doctoral theses). Secondary sources were also considered, such as bibliographic reviews and electronic databases (Dialnet, Google Scholar, Scopus, Teseo, and ProQuest). The search strategy within the databases was performed via the advanced search tool, restricting the date search from 2010 to 2020. The searched descriptors were also established hierarchically, starting from the most general to the most specific. The primary descriptors are as follows: emotional education, emotional intelligence, programme, and evaluation. Some widened combined descriptors led to a wide variety of results, depending on the combination of the terms from those previously mentioned.

One of the first-considered selection criteria from the data obtained was the published quasi-experimental research (with or without a control group), whose independent variable was linked to the implementation of specific programmes for EI development and its dimensions at any educational stage. The second well-considered selection criteria, emotional education programme/intervention, considered that it directly or indirectly develops some of the measurements or components integrated into the main EI models. To choose the most rigorous studies, the research focused on scientific quality manuscripts in terms of validity and reliability. In this sense, the internal validity was determined by the research design, the sample, and the instruments, where we observed that many articles did not describe the procedure for implementing and evaluating the defined programme.

Most of the studies presented use a triangulation of the data, using sources that make up the reality studied (students, families, and teachers), where the triangulation combines qualitative and quantitative techniques. Regarding the reliability of the assessment instruments, the studies that presented evidence collected through various assessment instruments and procedures in different groups and during different times were also considered as appropriate for this review. Thus, the study’s credibility is reinforced by the problem statement’s consistency regarding the specific situation it intends to address. The main exclusion criterion was the lack of scientific rigour, considering the studies’ validity and reliability.

## 6. Results

From the results obtained from the chosen studies, it can be observed that the type of theoretical orientations underlying some of the programmes is not described. Such an aspect can be observed in the review and coincides with the Evaluation Report of Emotional and Social Education and Life Skills programmes [12]. A large part of the intervention programmes has clear research goals, and the same researchers teach the activities integrated from the implemented programmes. Although there exist the hypotheses that behavioural and cognitive-behavioural-oriented programmes are more systematic and pose more explicit and more effective objectives, the meta-analyses seem to indicate that the most effective programmes are those that are theoretically coherent and highly interactive and use a variety of didactic methods through small groups and work–life skills, develop environmental strategies, and are integrated into the community [12]. Thus, Table 3 presents the characteristics of the programmes according to the educational stage in which they were developed, the place of implementation, the training of teachers, whether the internal or external personnel at the centre oversaw the intervention, as well as the participation of families, the number of lessons, and their duration (Table 3).

Regarding the educational level at which the selected interventions are developed, most of them are carried out in the stages of primary and secondary education. This information coincides with the meta-analysis obtained by Puertas-Molero et al. [22], highlighting that there is a lack of EE proposals evaluated in the infant education stage. It is also noteworthy that fewer EE programmes are systematised and assessed at Baccalaureate and University levels. In education environments, emotional imbalances are markedly manifested. Schools can be stressors in terms of, for example, competitiveness, grades, rivalries between peers, classroom participation, exams, homework, acceptance, fear of failure, and parental disappointments [23].

Regarding the nationality of the assessed programmes, the meta-analysis previously mentioned coincides with the fact that Spain is one of the countries with the most EE programme implementations in the last decade. According to Bisquerra [17], although the implementation of EE programmes is increasing, and there are increasingly more EE proposals in school settings, few systematised actions have been duly evaluated. In relation to the data collected regarding the teachers’ training and the format of the interventions, the results are very revealing of the fact that, in only 21.9% of the implementations, teachers have previously been trained in specific EE issues, near 50% are not trained, and, in 29.2% of the evaluations carried out, this information is not specified. Thus, only 11% of the teachers directly applied the programme’s activities, since, in 34.1%, they were directed and taught by non-school personnel (psychologists and university researchers). The meta-analysis obtained by Durlak and Weissberg [24] indicates that regular teachers from the educational centre taught 53% of the 213 selected programmes, 21% by external personnel and 26% by various agents, including the family.

Concerning the effectiveness of the programmes in which the family members, and internal and external personnel participated, the differences were insignificant. The Evaluation Report of Emotional and Social Education and Skills Programmes for Life [12] presents the analysis of 70 programmes, for which schoolteachers taught the majority. In more than half of the studies, they were the only trainers in direct contact with the students, while in 38% of the interventions, professionals were involved, such as psychologists or researchers. It is disappointing that, for the same report, in many cases, the characteristics (such as training and experience) of the instructors of the EE programmes and their relationship with the success or failure of the intervention are not described. Such a lack of information coincides with the results obtained for the present review, since, in almost 30% of the scrutinised programmes, it is not specified who takes responsibility for implementing the activities.

Apropos of the above, teacher training and socio-emotional competencies link students’ academic performance and socio-affective development [17]. Thus, emotionally competent teachers respond more effectively to their students’ needs by recognising and understanding their emotions, understanding the cognitive evaluations associated with those emotions and the resulting behaviours. In this way, teachers with high socio-emotional competence manage to prompt a better classroom climate, influencing student outcomes and teacher well-being [25]. Scientific evidence also shows that EE programmes result in significant improvements for teachers regarding the development of their emotional competence, where, at times, teachers experience more negative emotions than students [23]. According to Pérez Escoda et al. [26], teachers participating in EE programmes improve their emotional awareness, resulting in a more remarkable ability to identify emotions using a richer and more precise emotional vocabulary.

In many countries whose education systems already include EE as a priority, programmes are developed to stimulate school social and emotional learning—Collaborative for Academic, Social, and Emotional Learning (CASEL) in the U.S. or the Botín Foundation in Spain. Universities and teacher training centres also include activities on socio-emotional competencies in the initial training and the training catalogue for teacher professional development. Such is the relevance that the EE field is gaining popularity in Spain, and the Emotions Laboratory has been created within the Faculty of Psychology at the University of Málaga. Likewise, there are master’s degrees in Emotional Education offered by different Spanish universities.

Regarding the results obtained from families’ participation in the selected intervention programmes, only 17% participated in any planned activities. In 83% of the programmes, it is not stated whether the family intervenes at any phase of the programme or an initial meeting to offer consent for their children to participate in the U.S. programme. In this sense, it would be interesting for families to be aware of the benefits of EEs and develop their skills, thus positively affecting the family nucleus. For this reason, it is crucial that the school exerts an influence on the families and maintains a fluid relationship. The emotional training of the families would allow them to achieve a greater sense of emotional well-being and contribute to the improvement of their children [15]. Studies, such as those conducted by González-Pineda and Núñez [27], confirm that the involvement of families in educational centres produces positive effects on the relational dynamics between students, teachers, schools, and their surroundings.

Regarding the duration of the programmes, several meta-analyses carried out before 2010, such as that by Kraag et al. [28] and Durlak et al. [4], point out that programmes of a short duration or low intensity (no more than 8–10 lessons or 2 months long) often show considerably less or even insignificant effects. For programmes to be effective, they should be of a particular length or duration, probably between 3 and 6 months (weekly classes). Even so, if subsequent booster sessions are not performed, long-term results may not be obtained [12]. Of the EE programmes selected in the review, which reported the number of sessions developed, 14.6% had less than 8 lessons. The mean was 17.2, with only 12 programmes exceeding this number of lessons. In terms of the duration, the average length of the programmes was 5.5 months. The meta-analysis of Puertas-Molero et al. [22] highlights that the interventions of a medium duration (from 4 to 11 months) are those that report the best results; they are followed by those of a more extended period (1 to 2 years). The average effect of the programmes lasting less than three months is insufficient.

### An Evaluation of the Chosen EE Programmes

Regarding the evaluation of the programmes, the type of research design used is summarised below, which includes a longitudinal study with a long-term follow-up, sample size, and families’ participation in the evaluation process (Table 4). The selected studies were carried out in their entirety as quasi-experimental or pre-experimental research designs with a pre-test and post-test, with 83% of the investigations involving a control group. The study was longitudinal, with follow-up evaluations for 17% of the cases. Not having a follow-up evaluation is one of the limitations of almost 100% of the investigations. In many studies, the question arises concerning the stability of the observed results over time (e.g., the Dulcinea Programme of Cejudo [29]). All the programme evaluations report on the effects measured after conducting post-test tests; little information is reported for the long-term outcomes. Corresponding to the meta-analyses mentioned before 2006, it can be said that there is still a considerable shortage of research with follow-up evaluations of more than one year.

Other meta-analyses highlight the stability of the effects over time [30], producing the so-called *sleeping effect* [31], which means that the impact throughout the follow-up, six months or more after completion, are more relevant than in the post-test [12]. The long-term effects could also be related to programme implementation characteristics, namely, type, extent, the intensity of the program, the features of the groups, jointly with the contextual factors of the community or the school or with a combination of these factors [12]. When considering the sample size, it is necessary to highlight that the average was 331 participants and that 39% of the programmes were evaluated using a sample size of fewer than 100 participants. In general terms, it could be said that the larger the sample size, the more representative the results are for the population. In this sense, some selected studies expose the small sample size as a limitation (e.g., the INTEMO-UR Programme by Sigüenza-Marín et al. [32]).

The students’ families were considered in the studies’ evaluation process; however, a minority included them. Additionally, more than 80% of the evaluations did not consider the family context as a source of data or, at least, did not specify it. Considering some authors like Bisquerra et al. [15], the interpersonal relationships occurring within the family make it a continuum of emotions and the context in which conflict becomes inevitable. For this reason, the fact that research studies on EE programmes scarcely value the information reported by family members to evaluate the effects on the emotional development of the intervened students remains inconsistent.

In total, there are 75 variables associated with different theoretical models. Several evaluated factors reveal one of the drawbacks of the scientific evaluation of EI: the different operationalisation of the construct and its dynamism. According to various authors, theorists lack an agreement on what EI is and how it should be evaluated, so most studies present the methodological differences [7]. In the selected studies, the variables that were assessed more frequently are the following: skills, social relations/social competence, emotional regulation, emotional intelligence, and academic performance. Regarding the effects of the programmes, the Evaluation Report of Emotional and Social Education and Life Skills programmes [33] points out that, in a meta-analysis of 70 interventions, the main short-term impacts (up to 6 months) are found in emotional and social skills, attitudes towards oneself, prosocial behaviour, academic performance, and reduction in antisocial behaviour.

Some of the effects previously mentioned decrease substantially in the medium and long term, although not until they are negligible; however, reducing or preventing mental disorders would increase in the long term. Regarding the variables, such as ‘Academic Performance’, the literature indicates that its study raises numerous difficulties, since it is a multidimensional construct determined by many variables (e.g., intelligence, motivation, and personality), and which is influenced by numerous personal factors, family members, or schoolchildren [31]. As previously mentioned, the different conceptualisations of emotional competencies make their study and the understanding of them complex, which is why many instruments have been generated, producing a controversy regarding the best measures to evaluate the construct [7].

In the education field, three evaluative approaches to EI have been used: (1) classic measurement instruments based on questionnaires and self-reports performed by the students themselves; (2) evaluation measures of external observers based on questionnaires filled out by the student’s classmates or the teachers themselves; And (3) EI’s ability or performance measures on emotional tasks that students must solve. The studies mainly reviewed the use of traditional evaluation instruments, such as questionnaires, scales, and self-reports, which have been the most used measures in psychology. In fact, 87% of the reviewed studies’ instruments date back over 10 years (Table 5). In EE, advances are beginning to become a reality, and the most recent research proposes developing and implementing skills or execution tools. They are more innovative in their procedure and format, evaluating the style with which students solve some emotional issues by comparing their responses with predetermined and objective scoring criteria. Both measures can be complemented and used, depending on the purpose [7].

Several of the reviewed studies’ evaluation instruments have been repeatedly used, although any preponderance could not be declared. The most important ones in education are associated with EI theoretical models and translated into Spanish. In future research, we suggest authors opt for instruments that are more adequately adapted to each cultural context’s idiosyncrasy and evaluate neurocognitive functioning (attention, memory, and executive functions). In this sense, it would be necessary to implement new psychometric models [34]. Thus, outpatient assessment is making a solid comeback, driven by information and communication technologies. Thanks to this, mobile devices can study people’s emotions, feelings, thoughts, and psychological symptoms in their natural environment and daily life. This methodology would serve to evaluate psychological constructs from more dynamic, personalised, and contextualised micro-longitudinal and ecological perspectives [34].

## 7. Discussion and Conclusions

The results are clear from the data collected regarding teacher training at the centres participating in the programme and the format of the interventions. In only 21.9% of the implementations, the school’s teachers were previously trained in specific EE issues; almost 50% were not trained, and in 29.2% of the evaluations carried out, this information was not specified. Thus, only 11% of the teachers directly applied the programme activities, since 34.1% were directed and taught by personnel from outside the school (psychologists and university researchers).

The meta-analysis by Durlak et al. [24] highlights that 53% of the 213 selected programmes were taught by the regular teachers from the educational centre, 21% by external personnel, and 26% by various agents, including the family. Although it was hypothesised that the effectiveness of the programmes in which family members participated and internal and external personnel to the school would be more significant, the differences did not turn out to be significant. The Evaluation Report of Emotional and Social Education and Life Skills programmes [12] exposes the analysis of 70 programmes in which schoolteachers taught most of them. More than half of the studies were the only trainers in direct contact with the students, and for 38% of the programmes, there were professionals involved, such as psychologists or researchers. This same report classifies as disappointing the fact that, in many cases, the characteristics (training and experience) of the instructors of the EE programmes are not described, nor their relationship with the success or failure of the intervention. This factor coincides with the results of this review since, in almost 30% of the programmes, it is not even specified who takes responsibility for implementing the activities.

Concerning the above, there is evidence of the relationship between teacher training and socio-emotional skills with academic performance and the socio-affective development of students [15]. Emotionally competent teachers respond more effectively to the needs of their students by better recognising and understanding their emotions, understanding the cognitive evaluations associated with those emotions, and the behaviour that derives from them. This aspect influences student outcomes and teacher well-being, creating a positive feedback loop [25]. In this way, teachers with high socio-emotional competence manage to establish a positive bond with the students, manage the class better, and build a healthy classroom climate. Sometimes, teachers experience more negative emotions than students (Gil-Madrona and Martínez [35]). The study by Pérez Escoda et al. [36] concludes that teachers participating in an EE programme improve emotional awareness, which results in a more remarkable ability to identify emotions by having a richer and more precise emotional vocabulary. An increase in regulation strategies is also confirmed and, because of this, there is a spectacular increase in assertive responses. Thus, scientific evidence shows that EE programmes for teachers produce significant improvements in developing their emotional competence.

Regarding the results obtained from the participation of families in the selected intervention programmes, only 17% of family members participate in any of the programmed activities. In 83% of the programmes, it is not stated whether the family is involved in any phase of the programme or simply required for an initial meeting to offer consent for their children to be participants in the EE programme. Such essential authors in EE and Neuroeducation, such as Daniel Goleman, Francisco Mora, and Rafael Bisquerra, highlight in most of their works the influence of family as the backbone of emotional development, configuring itself as the first school and children’s primary role model. The students bring an evaluative style marked by the characteristics of their closest context. Therefore, the school needs to influence families and maintain fluid relationships. It would be interesting for families to know the benefits of EE and develop their skills, thus having a greater chance of positively affecting the family nucleus. Emotional training for mothers and fathers would allow them to achieve a greater sense of emotional well-being and contribute to that of their children [37].

The review of the 41 selected studies, based on EE programmes developed and evaluated in the last decade, allowed us to analyse and comprehend the characteristics of their implementation and the research activity around the interventions. Primary and secondary education are the stages in which the most significant number of programmes have been developed. However, the age of schoolchildren in infant education would be conducive to developing emotional competencies [17]. Baccalaureate and university’s stages should also have greater relevance, where social-emotional work would allow for more effective stress and anxiety management and a better academic performance at these higher educational levels [34]. Even though many investigations endorse EE’s benefits [20], only a small percentage of the total group of students is acknowledged in Spain. Thus, scarce EI programmes have been duly implemented and evaluated in the Spanish population [35]. In general, emotional training experiences tend to respond to sporadic programmes as a result of private initiatives, university personnel with research objectives, or eventual plans that try to respond to students’ emotional needs and are not attended by the formal educational system [34]. It would be necessary to evaluate the EE programmes carried out in schools and not assessed. Therefore, they remain unnoticed, and their effectiveness is not yet proven. In this sense, the disconnection between school reality and research activity is evident.

Considering the information derived from the review, it seems as if the studies existed because of the need to investigate from the side of universities and not for offering feedback to the educational agents. Therefore, a detection process for EE programmes is being carried out in the natural education settings and their link to universities or other promoters of scholarly research. Coinciding with the Evaluation Report of Emotional and Social Education and Life Skills programmes [33] and with the meta-analysis of [24], this review confirms that a large part of the EE programmes evaluated is taught by personnel outside the educational centre where the programme is being implemented. In the case of the teachers who develop the programme’s activities, they do not always have training in socio-emotional competencies, which is one of the fundamental pillars of success due to the teachers’ ability to influence, in different aspects, the students.

It is also shown that families’ participation is not relevant either in the development or evaluation of the programmes, despite being the main element influencing the emotional life in the early stages of life [37]. Regarding the duration of the programmes, it would be necessary that, after the 5.5 months obtained as an average of the selected interventions, other subsequent actions be added to reinforce learning and carry out longitudinal follow-up evaluations to assess long-term effects. Likewise, research would be necessary to offer data regarding the number and characteristics of the programmes’ sessions and activities to change the target population [38,39] positively. Regarding the research activity surrounding the evaluation of EE programmes, all of them start from the hypothesis that the intervention affects EI dimensions.

From the studies’ reviews, the referenced dependent variables are obtained, which result in significant differences. The multitude of results reveals the EI construct’s complexity and the need to make it operational and therefore more practical. The evaluation instruments used in the reviewed studies are related to the most relevant theoretical models of EI. Future research measures with self-reports, questionnaires, and traditional scales could complement the ability and other instruments based on psychometric models that can use new technologies to allow for evaluations in daily life contexts [32]. Regarding the studies’ external validity, it would be reinforced with the description of the implementation of the programmes and their evaluation process. Thus, they could be replicated in other school settings.

In summary, including a diagnosis before the intervention, and knowing what mediating variables influence its development, would facilitate an adaptation to the participants’ characteristics and the game’s context, increasing its effectiveness. The EE field requires a certain systematisation, considering the interventions and the results obtained to date. It would also be necessary to link schools with educational research entities, such as, for example, faculties of education to plan and evaluate programmes in the natural educational environment. It would also be essential to carry out complementary studies to determine flexible parameters that increase EE programmes’ effectiveness and make them adaptable to the characteristics of the settings in which they are to be developed, leading to generalisable procedures.

## Figures and Tables

**Table 1 children-09-00456-t001:** Synthesis of appointed studies (self-elaboration).

Programme	Stage	Authors	Place	Variables (sig.)
Programa de Desarrollo Psicoafectivo y Educación Emocional (Pisotón)	1	Manrique-Palacio, K. P., Zinke, L., and Russo, A. R. (2018)	CO	Addressing conflict, expressing emotions, and overcoming resistance
Programa de Educación Emocional de 3 a 6 años de Bisquerra	1	Román, N. S., Risoto, M. A., and Marín, A. H. (2018)	Murcia SP	Awareness, emotional regulation, self-esteem, socio-emotional skills, and life skills
Programa EMO-ACCIÓN	1	Serrano, A. C. (2015)	Castilla y León SP	Empathetic skills, conflict resolution, emotional regulation, and emotional identification
Proyecto educativo de Inteligencia Emocional en Infantil y Primaria	1, 2	Aguaded Gómez, M. C., and Pantoja Chaves, M. J. (2015)	Andalucía SP	Conflicts, regulation, expression of emotions, openness to dialogue, recognition of expressions, and identification of consequences of acts
Programa de intervención de Mindfulness	1	Almansa, G., Budía, M., López, J. L., Márquez, M., Martínez, A. I., Palacios, B. and José, E. (2014)	Andalucía SP	Emotional regulation, attention, school coexistence, attentional profile, personal growth, self-concept, and well-being
Programa de Educación Emocional a través de la Música (PEEM)	1	Arnau, J. P. (2019)	C. Valenciana SP	Physical self-concept, intellectual self-concept, anxiety, and academic performance
Programa EDI: ¿quieres viajar por el planeta de las emociones?	1	Bertomeu, R. B. (2016)	C. Valenciana SP	Violent behaviour, social and school adaptation, academic performance, and impulsive behaviour
Programa de reconocimiento de emociones	1	Celdrán Baños, J., and Ferrándiz García, C. (2012)	Murcia SP	Emotional perception and emotion recognition
Programa de Educación Emocional Happy 8–12	1	Cuenca, E. C., Escoda, N. P., Morente, A. R., and Guiu, G. F. (2019)	Cataluña SP	Regulation, anxiety, and academic performance
Programa de danza libre-creativa en Educación Física	1	Domínguez, C. L., and Castillo, E. (2017)	Andalucía SP	Self-knowledge, physical self-perception, body esteem, and emotional stability
Programa de Educación Emocional en el desarrollo de las competencias emocionales Grop	1	Filella-Guiu, G., Pérez-Escoda, N., Agulló-Morera, M. J., and Oriol-Granado, X. (2014)	México MX	Emotional intelligence, adaptability, and positive impression
Programa de Educación Emocional en Educación Primaria	1	Filella-Guiu, G., Pérez-Escoda, N., Agulló-Morera, M. J., and Oriol-Granado, X. (2014)	Cataluña SP	Emotional awareness, emotional regulation, emotional autonomy, social competence, life skills and well-being, interpersonal and intrapersonal components, adaptability, and positive impression
Intervención breve basada en Mindfulness	1	García Rubio, C., Luna Jarillo, T., Castillo Gualda, R., and Rodríguez Carvajal, R. (2016)	Madrid SP	Behaviour problems and academic performance
Programa para la enseñanza del balonmano con la estructura del modelo de Educación Deportiva	1	García-López, L. M., del Campo, D. G. D., González-Víllora, S., and Valenzuela, A. V. (2012)	Castilla-La Mancha SP	Violent behaviour, assertiveness, social skills, teamwork, and dynamic attitudes
Adaptaciones del Programa DIE (Desarrollando la Inteligencia Emocional) y del Programa Decide Tú	1	Hermosell, J. D. D. G., and Romero, I. M. M. (2011)	Extremadura SP Portugal PT	Knowledge and identification of good and not-so-good emotions, conflict avoidance and identification, and acceptance of one’s own and others’ characteristics
Programa de Mindfulness e Inteligencia Emocional (PINEP)	1	Luna-Pedrosa, J. F. (2017)	Andalucía SP	Academic performance, attention, processing speed, and mindfulness
Programa de Educación Emocional para Primaria	1	Merchán, I. M., Bermejo, M. L., and de Dios González, J. (2014)	Andalucía SP	Emotional intelligence, social relationships, self-awareness, self-control, emotional use, and empathy
Programa para el Bienestar y Aprendizaje Socio Emocional (BASE)	1	Milicic, N., Alcalay, L., Berger, C., and Álamos, P. (2013)	Chile CL	Self-perception of emotional skills, social skills, perception of the school climate, social integration, self-esteem, and perception of relationships with parents and teachers
Programa “VERA” de Educación Emocional	1	Moreno, S. C. B. (2017)	Madrid SP	Emotional regulation, recognising own emotions, leadership, cooperation, social adjustment, and pro-image
Programa “Educación Responsable” de la Fundación Botín	1	Palomera, R., Melero, M.A. and Briones, E (2018)	Cantabria SP	Social skills, emotional regulation, self-awareness, self-perception, creativity, prosocial behaviour, stress, mood, withdrawal, and intrapersonal and interpersonal competence
Devagar se vai ao longe (Lento pero seguro)	1	Raimundo, R., Marques-Pinto, A., and Lima, M. L. (2013)	Portugal PT	Emotional awareness, anxiety, peer relationships, self-control, and aggression
Programa RULER	1	Reyes, M. R., Brackett, M. A., Rivers, S. E., Elbertson, N. A., and Salovey, P. (2012)	United States US	Social skills, coexistence, and work habits
Inteligencia emocional en las clases de educación física	1	Ruiz, G., Lorenzo, L., and García, Á. (2013)	Madrid SP	Stress, sadness, emotional regulation, and worry
Programa de educación de las emociones: La Con-vivencia	1	Tur-Porcar, A., Mestre, V., Samper, P., Malonda, E., and Llorca, A. (2014)	C. Valenciana SP	Social skills, violent behaviour, regulation, coping, empathy, emotional stability, and prosocial behaviour
Proyecto VyVE (Vida y Valores en Educación)	1, 2	Zabal, M. Á. M., and Martín, R. P. (2011)	Cantabria SP	Clarity, positive parent relationships, reparation, social skills, prosocial behaviour, assertiveness, self-esteem, personal adjustment, and negative attitude toward school
Programa Educativo de Inteligencia Emocional (PEDIE)	2	Bailón, O. A. F., Peñaloza, J. L., Contreras, G. N., and Sierra, M. D. L. D. V. (2013)	México MX	Emotional perception and emotional facilitation skills
Programa AEdEm para Educación Secundaria	2	Calleja, L. S., Gómez, G. R., and Jiménez, E. G. (2018)	Andalucía SP	Awareness, emotional autonomy, life skills, and well-being
Programa INTEMO	2	Castillo, R., Salguero, J. M., Fernández-Berrocal, P., and Balluerka, N. (2013)	Andalucía SP	Violent behaviour, emotional regulation, and hostility
Programa de Intervención Psicopedagógica en Educación Emocional (PIPEE)	2	Cifuentes Sánchez, M. (2017)	Castilla-La Mancha SP	Total emotional intelligence, perception, understanding, regulation, and academic performance
Programa de Educación Emocional Happy 12–16	2	Cuenca, E. C., Escoda, N. P., Morente, A. R., and Guiu, G. F. (2019)	Cataluña SP	Emotional awareness, emotional autonomy, and academic performance
Programa de Inteligencia Emocional Intensivo (PIEI) en la inteligencia emocional y la conducta prosocial (CP)	2	García, G. A. R. (2018)	Madrid SP	Clarity, attention, repair, social skills, and prosocial behaviour
Programa de Inteligencia Emocional para adolescentes. INTEMO-UR	2	Marin, V. S., Guisado, R. C., Albéniz, A. P., and Fonseca-Pedrero, E. (2019)	La Rioja SP	Self-esteem, empathy, and self-perception of emotional abilities
Programa de Intervención en Inteligencia Emocional Plena (PINEP)	2	Ponce, N., and Aguaded, E. (2016)	Andalucía SP	Self-perception of emotional skills, emotional clarity, and emotional understanding
Programa “Dulcinea” de Educación Emocional	2	Cejudo, J. (2015)	Castilla-La Mancha SP	Violent behaviour, school performance, fluid and crystallised intelligence, emotional regulation, and hyperactivity
Programa Mínimo de Incremento Prosocial (PMIP)	2	Romersi, S., Fernández, J. R. M., and Roche, R. (2011)	Cataluña SP	Social skills and coexistence
Programa INTEMO	2	Ruiz-Aranda, D., Salguero, J. M., Cabello, R., Palomera, R., and Berrocal, P. F. (2012)	Andalucía SP	Emotional perception, regulation, emotional expression, depression, stress, and somatisation
Programa de inteligencia emocional en factores socioemocionales y síntomas psicosomáticos	2	Sarrionandia, A., and Garaigordobil, M. (2017)	País Vasco SP	Intrapersonal and interpersonal intelligence, mood, somatisation, happiness, emotional instability, and agreeableness
Programa de Educación Emocional para Adolescentes (PREDEMA)	2	Zegarra, S. P., Schoeps, K., Castilla, I. M., and Carbonell, A. E. (2019)	C. Valenciana SP	Emotional awareness, emotional regulation, emotional autonomy, and social competence
Programa de entrenamiento en conciencia plena (Mindfulness)	3	Justo, C. F., de la Fuente Arias, M., and Granados, M. S. (2011)	Andalucía SP	Self-concept, self-esteem, coping, and operability in the task
Programa de competencias emocionales	3	Martínez-Vilchis, R., Reynoso, T. M., and Rivera, J. P. (2017)	México MX	Victimisation, justification of cyberbullying, and perpetration
Programa de Inteligencia Emocional Plena (INEP)	4	Enríquez-Anchondo, H. A. (2011)	México MX	Emotional regulation, conscientiousness, empathy, emotional repair, extroversion, mindful planning, and putting into perspective
Programa Emociona’t	4	Fonseca-Pedrero, E., Pérez-Albéniz, A., Ortuño-Sierra, J., and Lucas-Molina, B. (2017)	La Rioja SP	Emotional awareness, emotional clarity, emotional repair, emotional understanding, and emotional management
Programa de Mindfulness e Inteligencia Emocional (PINEP)	2	Luna-Pedrosa, J. F. (2017)	Andalucía SP	Academic performance, attention, processing speed, and mindfulness
Programa de Educación Emocional para Primaria	2	Merchán, I. M., Bermejo, M. L., and de Dios González, J. (2014)	Andalucía SP	Emotional intelligence, social relationships, self-awareness, self-control, emotional use, and empathy
Programa para el Bienestar y Aprendizaje Socio Emocional (BASE)	2	Milicic, N., Alcalay, L., Berger, C., and Álamos, P. (2013)	Chile CL	Self-perception of emotional skills, social skills, perception of the school climate, social integration, self-esteem, and perception of relationships with parents and teachers
Programa “VERA” de Educación Emocional	2	Moreno, S. C. B. (2017)	Madrid SP	Emotional regulation, recognising own emotions, leadership, cooperation, social adjustment, and pro-image
Programa “Educación Responsable” de la Fundación Botín	2	Palomera, R., Melero, M.A., and Briones, E (2018)	Cantabria SP	Social skills, emotional regulation, self-awareness, self-perception, creativity, prosocial behaviour, stress, mood, withdrawal, and intrapersonal and interpersonal competence.
Devagar se vai ao longe (Lento pero seguro)	2	Raimundo, R., Marques-Pinto, A., and Lima, M. L. (2013)	Portugal PT	Emotional awareness, anxiety, peer relationships, self-control, and aggression
Programa RULER	2	Reyes, M. R., Brackett, M. A., Rivers, S. E., Elbertson, N. A., and Salovey, P. (2012)	United States U.S.	Social skills, coexistence, and work habits
Inteligencia emocional en las clases de educación física	2	Ruiz, G., Lorenzo, L., and García, Á. (2013)	Madrid SP	Stress, sadness, emotional regulation, and worry
Programa de educación de las emociones: La Con-vivencia	2	Tur-Porcar, A., Mestre, V., Samper, P., Malonda, E., and Llorca, A. (2014)	C. Valenciana SP	Social skills, violent behaviour, regulation, coping, empathy, emotional stability, and prosocial behaviour
Proyecto VyVE (Vida y Valores en Educación)	2, 3	Zabal, M. Á. M., and Martín, R. P. (2011)	Cantabria SP	Clarity, positive parent relationships, repair, social skills, prosocial behaviour, assertiveness, self-esteem, personal adjustment, and negative attitude towards school
Programa Educativo de Inteligencia Emocional (PEDIE)	3	Bailón, O. A. F., Peñaloza, J. L., Contreras, G. N., and Sierra, M. D. L. D. V. (2013)	México MX	Emotional perception and emotional facilitation skills
Programa AEdEm para Educación Secundaria	3	Calleja, L. S., Gómez, G. R., and Jiménez, E. G. (2018)	Andalucía SP	Awareness, emotional autonomy, life skills, and well-being
Programa INTEMO	3	Castillo, R., Salguero, J. M., Fernández-Berrocal, P., and Balluerka, N. (2013)	Andalucía SP	Violent behaviour, emotional regulation, and hostility
Programa de Intervención Psicopedagógica en Educación Emocional (PIPEE)	3	Cifuentes Sánchez, M. (2017)	Castilla-La Mancha SP	Total emotional intelligence, perception, understanding, regulation, and academic performance
Programa de Educación Emocional Happy 12–16	3	Cuenca, E. C., Escoda, N. P., Morente, A. R., and Guiu, G. F. (2019)	Cataluña SP	Emotional awareness, emotional autonomy, and academic performance
Programa de Inteligencia Emocional Intensivo (PIEI) en la inteligencia emocional y la conducta prosocial (CP)	3	García, G. A. R. (2018)	Madrid SP	Clarity, attention, repair, social skills, and prosocial behaviour
Programa de Inteligencia Emocional para adolescentes. INTEMO-UR	3	Marin, V. S., Guisado, R. C., Albéniz, A. P., and Fonseca-Pedrero, E. (2019)	La Rioja SP	Self-esteem, empathy, and self-perception of emotional abilities
Programa de Intervención en Inteligencia Emocional Plena (PINEP)	3	Ponce, N. and Aguaded, E. (2016)	Andalucía SP	Self-perception of emotional skills, emotional clarity, and emotional understanding
Programa “Dulcinea” de Educación Emocional	3	Cejudo, J. (2015)	Castilla-La Mancha SP	Violent behaviour, school performance, fluid and crystallised intelligence, emotional regulation, and hyperactivity
Programa Mínimo de Incremento Prosocial (PMIP)	3	Romersi, S., Fernández, J. R. M., and Roche, R. (2011)	Cataluña SP	Social skills and coexistence
Programa INTEMO	3	Ruiz-Aranda, D., Salguero, J. M., Cabello, R., Palomera, R., and Berrocal, P. F. (2012)	Andalucía SP	Emotional perception, regulation, emotional expression, depression, stress, and somatisation
Programa de inteligencia emocional en factores socioemocionales y síntomas psicosomáticos	3	Sarrionandia, A. and Garaigordobil, M. (2017)	País Vasco SP	Intrapersonal and interpersonal intelligence, mood, somatisation, happiness, emotional instability, and agreeableness
Programa de Educación Emocional para Adolescentes (PREDEMA)	3	Zegarra, S. P., Schoeps, K., Castilla, I. M., and Carbonell, A. E. (2019)	C. Valenciana SP	Emotional awareness, emotional regulation, emotional autonomy, and social competence
Programa de entrenamiento en conciencia plena (Mindfulness)	4	Justo, C. F., de la Fuente Arias, M., and Granados, M. S. (2011)	Andalucía SP	Self-concept, self-esteem, coping, and operability in the task
Programa de competencias emocionales	4	Martínez-Vilchis, R., Reynoso, T. M., and Rivera, J. P. (2017)	México MX	Victimisation, justification of cyberbullying, and perpetration
Programa de Inteligencia Emocional Plena (INEP)	5	Enríquez-Anchondo, H. A. (2011)	México MX	Emotional regulation, conscientiousness, empathy, emotional repair, extroversion, mindful planning, and putting into perspective
Programa Emociona’t	5	Fonseca-Pedrero, E., Pérez-Albéniz, A., Ortuño-Sierra, J., and Lucas-Molina, B. (2017)	La Rioja SP	Emotional awareness, emotional clarity, emotional repair, emotional understanding, and emotional management
Programa de Mindfulness e Inteligencia Emocional (PINEP)	2	Luna-Pedrosa, J. F. (2017)	Andalucía SP	Academic performance, attention, processing speed, and mindfulness
Programa de Educación Emocional para Primaria	2	Merchán, I. M., Bermejo, M. L., and de Dios González, J. (2014)	Andalucía SP	Emotional intelligence, social relationships, self-awareness, self-control, emotional use, and empathy
Programa para el Bienestar y Aprendizaje Socio Emocional (BASE)	2	Milicic, N., Alcalay, L., Berger, C., and Álamos, P. (2013)	Chile CL	Self-perception of emotional skills, social skills, perception of the school climate, social integration, self-esteem, and perception of relationships with parents and teachers
Programa “VERA” de Educación Emocional	2	Moreno, S. C. B. (2017)	Madrid SP	Emotional regulation, recognising own emotions, leadership, cooperation, social adjustment, and pro-image
Programa “Educación Responsable” de la Fundación Botín	2	Palomera, R., Melero, M.A., and Briones, E (2018)	Cantabria SP	Social skills, emotional regulation, self-awareness, self-perception, creativity, prosocial behaviour, stress, mood, withdrawal, and intrapersonal and interpersonal competence
Devagar se vai ao longe (Lento pero seguro)	2	Raimundo, R., Marques-Pinto, A., and Lima, M. L. (2013)	Portugal PT	Emotional awareness, anxiety, peer relationships, self-control, and aggression
Programa RULER	2	Reyes, M. R., Brackett, M. A., Rivers, S. E., Elbertson, N. A., and Salovey, P. (2012)	United States US	Social skills, coexistence, and work habits
Inteligencia emocional en las clases de educación física	2	Ruiz, G., Lorenzo, L., and García, Á. (2013)	Madrid SP	Stress, sadness, emotional regulation, and worry
Programa de educación de las emociones: La Con-vivencia	2	Tur-Porcar, A., Mestre, V., Samper, P., Malonda, E., and Llorca, A. (2014)	C. Valenciana SP	Social skills, violent behaviour, regulation, coping, empathy, emotional stability, and prosocial behaviour
Proyecto VyVE (Vida y Valores en Educación)	2, 3	Zabal, M. Á. M. and Martín, R. P. (2011)	Cantabria SP	Clarity, positive parent relationships, repair, social skills, prosocial behaviour, assertiveness, self-esteem, personal adjustment, and negative attitude towards school
Programa Educativo de Inteligencia Emocional (PEDIE)	3	Bailón, O. A. F., Peñaloza, J. L., Contreras, G. N., and Sierra, M. D. L. D. V. (2013)	México MX	Emotional perception and emotional facilitation skills
Programa AEdEm para Educación Secundaria	3	Calleja, L. S., Gómez, G. R., and Jiménez, E. G. (2018)	Andalucía SP	Awareness, emotional autonomy, life skills, and well-being
Programa INTEMO	3	Castillo, R., Salguero, J. M., Fernández-Berrocal, P., and Balluerka, N. (2013)	Andalucía SP	Violent behaviour, emotional regulation, and hostility
Programa de Intervención Psicopedagógica en Educación Emocional (PIPEE)	3	Cifuentes Sánchez, M. (2017)	Castilla-La Mancha SP	Total emotional intelligence, perception, understanding, regulation, and academic performance
Programa de Educación Emocional Happy 12–16	3	Cuenca, E. C., Escoda, N. P., Morente, A. R., and Guiu, G. F. (2019)	Cataluña SP	Emotional awareness, emotional autonomy, and academic performance
Programa de Inteligencia Emocional Intensivo (PIEI) en la inteligencia emocional y la conducta prosocial (CP)	3	García, G. A. R. (2018)	Madrid SP	Clarity, attention, repair, social skills, and prosocial behaviour
Programa de Inteligencia Emocional para adolescentes. INTEMO-UR	3	Marín, V. S., Guisado, R. C., Albéniz, A. P., and Fonseca-Pedrero, E. (2019)	La Rioja SP	Self-esteem, empathy, and self-perception of emotional abilities
Programa de Intervención en Inteligencia Emocional Plena (PINEP)	3	Ponce, N. and Aguaded, E. (2016)	Andalucía SP	Self-perception of emotional skills, emotional clarity, and emotional understanding

Note. Infant education (1); primary education (2); compulsory secondary education (3); baccalaureate education (4); and university education (5).

**Table 2 children-09-00456-t002:** Emotional competencies and components (self-elaboration).

Authors	Competencies	Skills
Gardner (1983)	Intrapersonal emotional intelligence	Understanding self-emotions; naming emotions; guiding self-behaviour.
	Interpersonal emotional intelligence	Understanding others; considering differences in moods, temperaments, motivations, and abilities; communicating; taking on various roles within groups.
Mayer and Salovey (1997)	Perception Emotional expression Assimilation and emotional analysis Regulación reflexiva de las emociones	
Goleman (1998)	Self-awareness	Emotional awareness; self-esteem; self-confidence.
	Self-regulation	Self-control; ability and integrity; innovation and adaptability.
	Motivation	Achieving motivation; commitment; initiative and optimism.
	Empathy	Understanding others; development of others; serviceability orientation; diversity learning; political awareness.
	Social skills	Influence; communication; conflict management; leadership; catalysts of change; establish links; collaboration and cooperation; team capabilities.
CASEL Program (2003)	Self-awareness	Identification of emotions; recognition of strengths.
	Self-management	Emotional management; goal setting.
	Social conscience	Understanding others; differences as a source of enrichment.
	Relationship skills	Communication; relationship building; negotiation; ability to say no.
	Responsible decision making	Situation analysis; assumption of responsibilities; respect for others; problem resolution.
Bar-On (2006)	Intrapersonal	To be accurately perceived, understood, and accepted; emotional self-awareness; assertiveness; emotional independence; self-realisation.
	Stress management	Stress tolerance; impulse control.
	Interpersonal	Empathy; social responsibility; inter-relationships.
	Adaptability	Adjustment to reality; flexibility; problem resolution.
	Frame of mind	Optimism; happiness.
Bisquerra (2010)	Emotional awareness	To be aware of self-emotions; identify and name emotions; understanding the emotions of others; become aware of the interaction between emotions, cognition, and behaviour.
	Emotional regulation	Appropriate emotional expression; regulation of emotions and feelings. Coping skills; competence to self-generate positive emotions.
	Emotional autonomy	Self-esteem; self-motivation; emotional self-efficacy; responsibility; positive attitude; critical analysis of social norms; resilience.
	Social competence	Mastery of basic social skills; respect for others; practice receptive communication; practice expressive communication; share emotions; prosocial behaviour and cooperation; assertiveness, prevention, and conflict resolution; ability to manage emotional situations.
	Life skills and wellness	Set adaptive goals; decision making; find help and resources; active and participatory, critical, responsible, and committed citizenship; emotional well-being; flow.

**Table 3 children-09-00456-t003:** Characteristics of the implemented EE programmes (self-elaboration).

	No. of Programmes	Percentages
**Stage**		
Infant	3	7.3
Primary	19	46.3
Secondary	13	31.7
Baccalaureate	2	4.8
University	2	4.8
Other	2	4.8
**Nationality**		
National (ES)	32	78
International (CL, CO, MX, PT, U.S.)		
National–International (ES-PT)		
**Teacher training**		
Yes	9	21.9
No	20	48.7
Unspecified	12	29.2
**Intervention formats**		
Centre’s teaching staff	11	26.8
External teaching staff	14	34.1
External teaching/intervention	4	9.7
Unspecified	12	29.2
**Family engagement**		
Participate in the programme	7	17
Do not participate in the programme; unspecified; only aware of the intervention	34	83
**Programme duration**		
Avg. no. of lessons = 17.2		
Avg. no. of months = 5.5		

**Table 4 children-09-00456-t004:** Characteristics of the evaluated EE programmes (self-elaboration).

	No. of Programmes	Percentages
Research design		
Quasi-experimental (pre-/post-test). Experimental and control groups? YES	34	83
Quasi-experimental (pre-/post-test). Experimental and control groups? NO	7	17
**Follow-up of the assigned programme and/or longitudinal study**		
Yes	7	17
No	34	83
**Sample size (Avg. 331 *)**		
<100	16	39
100 to 200	7	17
201 to 300	5	12.1
>300	13	31.7
**Families in the evaluation**		
Participate in the evaluation	8	19.5
Unspecified; do not participate; participate in their consent; or just the socioeconomic family aspects are considered	33	80.5

* Windsor average, eliminating upper/lower values equating to the closest ones.

**Table 5 children-09-00456-t005:** No. of evaluation instruments by decades (self-elaboration).

	No. of Evaluation Instruments	Percentages
Year		
Before 1980	34	83
1981 to 1989	16	39
1990 to 1999	7	17
2000 to 2009	5	12.1
2010 to 2020	33	80.5

## Data Availability

No applicable.

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
