# Peer review of "The Development of Emotional Programmes in Education Settings during the Last Decade"

_children, 2022, doi:10.3390/children9040456_

Round 1

Reviewer 1 Report

The manuscript presents a review on literature on programmes on emotional education on various educational levels with a focus on the study characteristics, the programmes‘ evaluation, and findings. First, the main models and research paradigms of emotional intelligence (EI) and the proposedly positive outcomes of high EI in students as identified in existing research are described. Consequently, the proposedly positive outcomes of emotion educational programms are listed. After the review’s objectives are described, the research methods and results are described. According tot he authors, results showed a need to integrate families into programme interventions, to train teachers in socio-emotional competencies and to connecting every-day educational with research activities.

Undoubtedly, a systematic overview of existing socio-emotional learning programmes in school or any other settings is valuable for their conceptualization, development and long-term implementation regarding results from evaluation studies. The manuscript therefore represents a relevant step to advance this line of research and to bring together emprirical research and the necessities of applied contexts such as educational contexts.

However, in ordert to let future research, particularly from outside Spanish speaking countries, benefit from this review, some major requirements should be acknowledged:

  1. One very general point concerning a revision should be a more systematic presentation of the reviewed studies so that also English speaking researchers can benefit from this overview (all reviewed studies are in Spanish language). The overview should include: a) a list or table of the reviewed studies including authors, title and publication form, a classification into the relevant theoretical framework of emotional skills, some of the other relevant characteristics as given in Table 1 and a short description oft he main results.
  2. At the same time, the character of a review would require that the level of interpretation of the results goes beyond the single studies, which is – at least in part - not the case in the present manuscript. In other words, based on a systematic presentation (see point 1) of the included studies, what can we learn analyzing the pattern of results considering the study characteristics? Moreover, the interpretation should become reproducable by the reader by having all necessary information in the manuscript.
  3. In order to have the information clear and available when showing the study characteristics, the various theoretical frameworks or models of emotional intelligence or emotional skills should be described a bit more in detail (ability vs. mixed models). The resulting derivation of consequences of having more or less emotional abilities or skills do not get clear based on such a short description. The theoretical relation of predictor (EI or emotional skills and whoch domain or facet of them) and possible outcomes (e.g., academic performance) should get very clear tot he reader. For example, the second half of the section 1.2 presents the outcomes of emitional education programmes but doesn’t assign them to the assumed effect of the related specific predictors. Where do the redictors come from (i.e. which model or theory) and how do they affect the outcomes.
  4. In this respect, the labels for certain concepts should used stringently throughout the manuscript. For example, the term capacity is used in a double meaning (i.e., lack of capacity as a lack of cognitive abilities and EI is also addressed as a capacity). Moreover, capacity is a rather unprecise concept compared to the concepts of abilities, skills, and traits and I would recommend to rather use them according to their precise conceptualization (e.g. as described in Weis & Conzelmann, 2015).

Minor aspect:

- Abbreviations should be introduced (i.e., EI, IE, EE)

Reference

Weis, S. & Conzelmann, K. (2015). Social Intelligence and Competencies. In: James D. Wright (Ed.), International Encyclopedia of the Social & Behavioral Sciences, 2nd edition, Vol 22. Oxford: Elsevier. pp. 371–379.

Author Response

Dear reviewer,

Please, find attached the revised manuscript with the suggested changes (red colour).

Best regards,

Authors

Reviewer 2 Report

The paper is well written and it reveals very good information about emotion education. I just have some suggestions which may come helpful to you.

Some more details should be addressed, such as:

The importance of emotion and its components.

What is emotion education? and the importance of it in the educational and clinical settings

What programs are considered in this study? Student emotion training, teacher emotion training, family emotion training, or combination of them? More explanation about their characteristics, and the importance of each program is needed.

Some more search could be done by Pub-Med, Since direct, ERIC,…

It would be more helpful if the results for students training program, teacher training program, and family training programs are in a separate table too.

Most of the discussion is about the students programs and its evaluation. Teacher and family programs and their evaluation should be discussed.

Author Response

Dear reviewer,

Please, find attached the revised manuscript with the suggested changes (blue colour).

Best regards,

Authors

Round 2

Reviewer 1 Report

I want to thank the authors for the revision of their manuscript which has addressed many of my points from the first review. Some aspects remain which are as follows (along the original points from review 1):

  1. systematic presentation of the reviewed studies: The authors added a Table with all reviewed studies including title of the program, authors, addressed educational level, place and trained variables. I recommended that the authors incorporate into the table also the characteristics that are addressed in the results section which they didn't so far (e.g. degree of interactivity and more importantly, a short description of the main results); see also point 2 and 3 in the last and this review.
  2. reproducability of interpretation: The description of results cannot entirely be followed by the reader. There are still elements in the results section that just describe other existing research (e.g. The Evaluation Report of Emotional and Social Education and Skills Programmes for Life) which does not belong in the results section of a study, but either in the theoretical sections or the discussion. At the same time, since the new Table 1 does not include all variables that are relevant for interpreting the results and more importantly, a short description of results, the given interpretations cannot be reproduced by the reader.
  3. presentation of the theoretical frameworks: The authors added a Table (Table 2) that gives an overview over the main EI models. 
  4. clarity of concepts: I would recommend some small changes to this Table 2. The label of column 2 should rather be "competenciy or intelligence domain", column 3 could then be titled "skills / abilites". Consequently, the four domains of the Mayer-Salovey model should be moved to column 3, in column 2 in this line could be included "emotional intelligence". If possible (as an optional complementation), new Table 1 could make a link to the underlying theoretical model as displayed in new Table 2, since one interpretation in the results section was, that the more theoretically coherent the program was, the more effective was the program.

Minor points: The abbreviation EE is not in brackets behind the first use of the relevant expression (emotion education).

Author Response

See files
